# Impacts of Drought and Climatic Factors on Vegetation Dynamics in the Yellow River Basin and Yangtze River Basin, China

Weixia Jiang [1,2], Zigeng Niu [1,2,*], Lunche Wang [1,2], Rui Yao [1,2], Xuan Gui [1,2], Feifei Xiang [1,2] and Yuxi Ji [3]

1   Hunan Key Laboratory of Remote Sensing of Ecological Environment in Dongting Lake Area,
    School of Geography and Information Engineering, China University of Geosciences, Wuhan 430079, China;
    jiangwx@cug.edu.cn (W.J.); wang@cug.edu.cn (L.W.); yaorui123@cug.edu.cn (R.Y.);
    guixuan@cug.edu.cn (X.G.); feifeixiang@cug.edu.cn (F.X.)
2   Key Laboratory of Regional Ecology and Environmental Change, School of Geography and Information
    Engineering, China University of Geosciences, Wuhan 430079, China
3   Wuhan Geomatics Institute, Wuhan 430022, China; jiyuxi_ss@163.com
*   Correspondence: nzg@cug.edu.cn; Tel.: +86-180-6412-7981

**Abstract:** Understanding the impacts of drought and climate change on vegetation dynamics is of great significance in terms of formulating vegetation management strategies and predicting future vegetation growth. In this study, Pearson correlation analysis was used to investigate the correlations between drought, climatic factors and vegetation conditions, and linear regression analysis was adopted to investigate the time-lag and time-accumulation effects of climatic factors on vegetation coverage based on the standardized evapotranspiration deficit index (SEDI), normalized difference vegetation index (NDVI), and gridded meteorological dataset in the Yellow River Basin (YLRB) and Yangtze River Basin (YTRB), China. The results showed that (1) the SEDI in the YLRB showed no significant change over time and space during the growing season from 1982 to 2015, whereas it increased significantly in the YTRB (slope = 0.013/year, $p < 0.01$), and more than 40% of the area showed a significant trend of wetness. The NDVI of the two basins, YLRB and YTRB, increased significantly at rate of 0.011/decade and 0.016/decade, respectively ($p < 0.01$). (2) Drought had a significant impact on vegetation in 49% of the YLRB area, which was mainly located in the northern region. In the YTRB, the area significantly affected by drought accounted for 21% of the total area, which was mainly distributed in the Sichuan Basin. (3) In the YLRB, both temperature and precipitation generally had a one-month accumulated effect on vegetation conditions, while in the YTRB, temperature was the major factor leading to changes in vegetation. In most of the area of the YTRB, the effect of temperature on vegetation was also a one-month accumulated effect, but there was no time effect in the Sichuan Basin. Considering the time effects, the contribution of climatic factors to vegetation change in the YLRB and YTRB was 76.7% and 63.2%, respectively. The explanatory power of different vegetation types in the two basins both increased by 2% to 6%. The time-accumulation effect of climatic factors had a stronger explanatory power for vegetation growth than the time-lag effect.

**Keywords:** drought; vegetation growth; Yellow River Basin; Yangtze River Basin; time-lag effects; time-accumulation effects

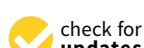



## 1. Introduction

Vegetation plays an indispensable role in regulating the carbon cycle, climate change, and the energy exchange between the atmosphere, land surface, and hydrological processes [1,2]. Large-scale vegetation cover changes (degradation and restoration) are important indicators for quantifying the impacts of natural evolution and human activities on the ecological environment [3,4]. Therefore, monitoring vegetation dynamics and

quantifying the impact of climate change on vegetation growth have become a hot topic in global change research, and it is of great significance for understanding the behavioral mechanisms of vegetation ecosystems.

Global warming has caused extreme weather and climate events to occur more frequently [5], and has particularly increased the intensity and frequency of drought events [6,7]. Drought can inhibit the normal growth of vegetation, and continuous drought can cause vegetation to die owing to lack of water, thereby significantly reducing vegetation productivity and causing regional vegetation extinction events [8]. Numerous studies have investigated the relationship between drought and vegetation growth [9–11]. For example, 152 drought events lasting more than three months in mainland China from 1982 to 2015 were identified using the three-dimensional clustering algorithm, and during long-term and severe drought events, the net primary productivity showed a significant decline [12]. Many studies have proved that climate variables such as temperature (TEM) and precipitation (PRE) are the main causes of drought [13–15]. Sun and Ma (2015) [16] concluded that the increase in TEM and the decrease in PRE showed a tendency to exacerbate drought in the Loess Plateau. Similarly, Yang et al. (2020) [15] proved that agricultural drought has a strong correlation with TEM and PRE.

TEM and PRE are the two major climatic factors that cause vegetation changes. Vegetation growth responds to the climate only when the climate change exceeds the tolerance range of the vegetation [17]. Therefore, the time-lag effect should be fully considered when exploring the climate–vegetation interaction mechanism. In recent years, increasing studies have shown that the response of vegetation to climate has a certain time-lag effect [18–20]. Gu et al. (2018) [21] found that vegetation exhibited different time-lag responses to different climate factors, and they also pointed out that it is necessary to further consider the impact of previously accumulated monthly TEM and PRE, i.e., the time-accumulation effect. Because the standardized precipitation evapotranspiration index (SPEI) can explain the accumulated water shortage or surplus [22], most previous studies used the SPEI at different time scales to quantify the accumulation effect [11,23]. However, the time-lag and time-accumulation effects of climatic factors on vegetation usually coexist [24,25], and these two effects were rarely investigated together in previous studies, or the two effects were considered separately without determining the best solution for each grid [20,26]. Therefore, in order to better understand the interaction between climate change and vegetation growth and to manage vegetation more effectively and protect vegetation from drought, more attention should be paid to the time-lag and time-accumulation effects of the main climatic driving forces on vegetation growth [27].

The Yellow River Basin (YLRB) and the Yangtze River Basin (YTRB) are important population settlements and water supply sources in China [28]. Droughts have occurred frequently in the two river basins in recent decades, and the duration and intensity of droughts have gradually increased, thereby adversely affecting water resources, the ecological environment, and socioeconomic systems [29,30]. Besides, the vegetation coverage in the YLRB and YTRB is an important ecological barrier that is indispensable for maintaining the ecological balance in China, neighboring regions, and the world [31–33]. After many large-scale ecological projects were implemented, significant changes have taken place in the vegetation coverage in the two basins [34,35]. The effects of drought and climate on vegetation have attracted widespread attention [35,36]. Jiang et al. (2019) [10] used the Pearson correlation analysis to study the impact of multi-time-scale drought on different vegetation types in the YTRB, and concluded that short-term and medium-term droughts have a greater impact on vegetation. In addition, Zhang et al. (2020) [35] analyzed the relationship between vegetation coverage and climate change at different time scales in the YLRB and YTRB based on the ensemble empirical mode decomposition method and showed that the response of vegetation to climate change becomes more prominent as the time scale increases. However, the time-lag effect and time-accumulation effect of climatic factors on the vegetation in the two basins have been less studied. It is necessary to understand the

impact of drought on vegetation growth and the response of vegetation activities to climatic changes in the YLRB and YTRB via continuous long-term data series.

This research aims to address the urgent need to analyze the impacts of drought and climate change on vegetation in the YLRB and YTRB. First, the temporal and spatial characteristics and changing trends of drought and vegetation coverage in the growing season (April to October) from 1982 to 2015 were investigated. The relationships among drought, climatic factors, and vegetation were then estimated on the pixel scale, and different time effect scenarios (time-lag effect, time-accumulation effect, combined effect, and no time effect) were considered to assess the dependence of vegetation on climatic factors. Finally, based on the determined time effect, a multiple linear regression model was established to quantitatively analyze the relationship between changes in both climatic factors and vegetation growth. The impact of human activities on vegetation growth via residual trends was also preliminarily examined.

## 2. Materials and Methods

### 2.1. Study Area

The Yellow River originates from the Qinghai–Tibet Plateau, flows through the Inner Mongolia Plateau, the Loess Plateau, and the Huang-Huai-Hai Plain, and finally discharges into the Pacific Ocean (Figure 1a). The YLRB is located at 96°E–119°E, 32°N–42°N, with a total length of approximately 5464 km and a drainage area of about $7.95 \times 10^5$ km$^2$ [37]. The average annual PRE ranges from 123 mm to 1021 mm, increasing from northwest to southeast, and the average annual TEM varies from about −4 °C to 14 °C [29]. The YLRB is mainly composed of arid and semiarid environments and is occupied by grasslands and croplands. The Yangtze River also originates in the Qinghai–Tibet Plateau and flows through the Yunnan–Guizhou Plateau, Sichuan Basin, and Jiangnan hills (Figure 1a). The YZRB is located at 24°30′N–35°45′N, 90°33′E–122°25′E [38]. It has a total length of approximately 6300 km and an area of about 1.8 million km$^2$, accounting for 18.8% of China's total land area. The basin is characterized by a typical subtropical monsoon climate, with a clear downward trend in TEM and PRE along the southeast–northwest direction [30,39]. The source area and upstream vegetation are dominated by alpine meadows and natural grasslands, the main vegetation type in the middle reaches is forest, and croplands are widely distributed in the middle and lower reaches of the plain and Sichuan Basin (Figure 1b).

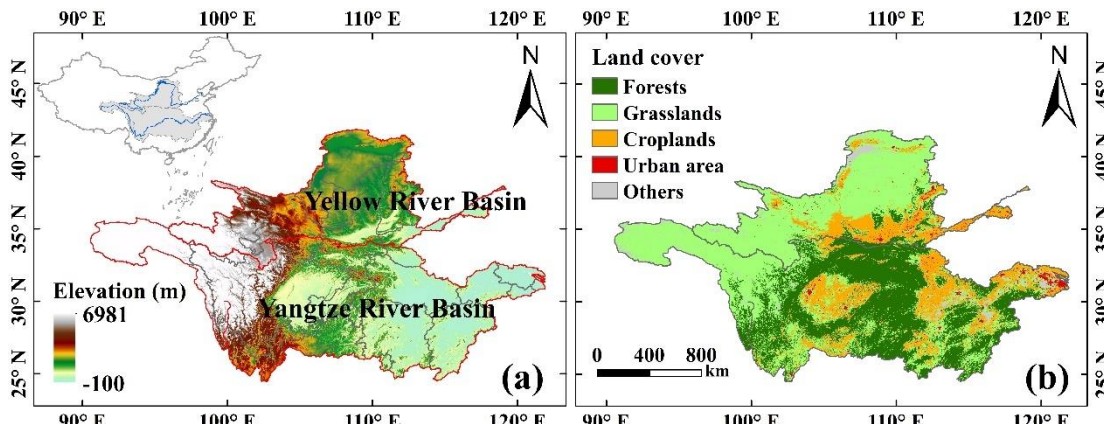

**Figure 1.** Location, elevation (**a**), and land cover types (**b**) of the Yellow River Basin and Yangtze River Basin.

### 2.2. Datasets

The standardized evapotranspiration deficit index (SEDI), a new drought index proposed by [40,41], quantifies the severity of drought based on the difference between actual evapotranspiration and the atmospheric evaporation requirement. The index has a spatial resolution of 0.25° × 0.25° and spans from 1982 to 2015. Compared with other drought

indexes based on PRE and TEM, the SEDI can reasonably detect droughts and wet–dry climate transitions on a monthly scale, and can also reproduce long-term trends [42]. In addition, the SEDI can capture the biological changes in the ecosystem in response to the dynamics of drought intensity more sensitively and highlight the signals of biological effects in drought; thus, it is more suitable for studying the impact of drought on vegetation [12]. The classifications of the dryness–wetness grade based on the SEDI are shown in Table 1. The SPEI and Self-Calibrated Palmer Drought Index (scPDSI) are traditional drought indices, and they were downloaded from http://digital.csic.es/handle/10261/153475 and https://crudata.uea.ac.uk/cru/data/drought, respectively (accessed on 15 January 2022). The 1-month, 3-month, 6-month, 12-month and 24-month scales of the SPEI (SPEI01, SPEI03, SPEI06, SPEI12, and SPEI24) and scPDSI were selected to explore the applicability of the SEDI in the YLRB and YTRB.

**Table 1.** Drought severity classification of SEDI values [12].

| SEDI | Classification |
|---|---|
| Less than −2.0 | Extreme drought |
| −1.99 to −1.5 | Severe drought |
| −1.49 to −1.0 | Moderate drought |
| −0.99 to −0.5 | Mild drought |
| −0.5 to 0.5 | Normal |
| 0.5 to 0.99 | Mildly wet |
| 1.0 to 1.49 | Moderately wet |
| 1.5 to 1.99 | Severely wet |
| Larger than 2.0 | Extremely wet |

The Normalized Difference Vegetation Index (NDVI) is a direct measure of the radiation absorbed by the canopy, which is dimensionless (ranging from 0 to 1) [43]. The NDVI has become the most widely used vegetation evaluation index due to its long history, simplicity, and dependence on easily obtained multi-spectral bands [44]. It is closely correlated with a series of interrelated biomass variables such as the leaf area index [45], vegetation cover [46], and green biomass [47]. The NDVI is an effective tool for coupling climate and vegetation distribution and performance at large spatio-temporal scales [48]. NDVI analysis of contemporary vegetation dynamics is particularly effective for large transboundary geographical regions with varied terrain, diverse vegetation, and diverse land-use types in the Northern Hemisphere [4]. The NDVI datasets obtained from the Global Inventory Modeling and Mapping Study (GIMMS 3 g) were used as the vegetation index. This dataset has the longest time series, so that it has been widely used in global vegetation monitoring. The dataset spans from 1982 to 2015, with a spatial resolution of 1/12° and an interval of 15 days.

The monthly PRE and TEM of the $0.25° \times 0.25°$ meteorological dataset from 1982 to 2015 (CN05.1) were used [49,50]. The CN05.1 dataset was constructed based on more than 2400 sites in China through the anomaly method in the interpolation process [51]. In this method, the daily grid climate and its anomalies were calculated using thin-plate smoothing splines, then anomalies were added to the climate, and the monthly average TEM and accumulated PRE were calculated to obtain the final dataset [52].

The land-use data were derived from the MODIS land cover type product (MCD12Q1) in 2015, which has a spatial resolution of 500 m. To maintain the consistency of the spatial resolution, the land cover data and NDVI data were resampled to match the spatial resolution of 0.25° using the nearest neighbor method [53].

*2.3. Methods*

2.3.1. Theil–Sen Median Trend and Mann–Kendall Test

Sen's slope and the Mann–Kendall (MK) trend test were applied to investigate the changing trends of drought, vegetation, and climatic factors during the period of 1982–2015. Sen's slope has high computational efficiency and allows missing values; therefore, it has been used in trend analysis of long-term series data [54,55]. Sen's slope is usually used in combination with the MK trend test [56,57], which is a method to measure the trend significance of sample time series. It has been widely used to detect trends in hydrological and meteorological time series [10,58].

2.3.2. Pearson Correlation Analysis

Pearson correlation analysis is a widely used method in statistics and has been widely used in previous studies [11,26,59,60]. In this study, this method was used to examine the correlation between the SEDI and NDVI during the growing season, as well as between climatic factors and the NDVI.

2.3.3. Temporal Effects of Climatic Factors on NDVI

Linear regression was applied to estimate the temporal effects of climatic factors on vegetation. Since time effects on a monthly scale are usually shorter than three months [27,61,62], time-lag and time-accumulation effects of up to four months were considered in this study.

$$\text{NDVI}_t = b \times \sum_{j=0}^{k} X_{t-i-j} + a \tag{1}$$

where $a$ and $b$ are regression coefficients, $X$ represents climate factors (TEM or PRE); the value range of $i$ and $k$ is 0 to 3 (i.e., 0 means no time effect, and 1–3 means a lag or accumulation of 1–3 months, respectively). A detailed introduction to this method can be obtained by referring to Ding et al. (2020) [61].

2.3.4. Multiple Linear Regression

A multiple linear regression model of the NDVI and two climatic factors was established to quantify the contribution of climatic factors to vegetation change.

$$\text{NDVI} = A \times \text{TEM} + B \times \text{PRE} + C \tag{2}$$

where A and B are regression coefficients and C is the error term. TEM and PRE are the adjusted time series according to the best time effects identified in Section 2.3.3.

2.3.5. Residual Trend Method

The residual trend method is a preferable method to distinguish the impacts of human activities and climate change on vegetation dynamics at a large spatial scale [43,63]. Using a regression model with climate factors as the dependent variable, the residuals between the observed NDVI and predicted NDVI were obtained [4]. The effect of artificial activities on the NDVI could be expressed by the trend change of residuals, which was calculated using Sen's slope method (Section 2.3.1). A positive trend of the residuals indicated that vegetation restoration was mainly caused by human influence, whereas a negative trend indicates that human activities led to vegetation degradation [31].

**3. Results**

*3.1. Spatiotemporal Variabilities of Drought, Climatic Factors, and Vegetation Coverage*

From 1982 to 2015, the average SEDI in the growing season of the YTRB showed a significant growth trend with a rate of 0.013/year ($p < 0.01$), indicating that drought was alleviated to a certain extent and the climate would be more humid in the future (Figure 2a). The growth rate of the SEDI in the YLRB was 0.002/year ($p > 0.01$). Compared with the

YTRB, the YLRB experienced more drought events during the study period. For example, the SEDI was lower than −0.5 in 1987, 1997, and 2000. The YLRB is located in the temperate continental and temperate monsoon climate zone, and drought events occur frequently due to the lack of PRE [64]. In addition, complex human activities have also become an important reason for the formation of drought in the YLRB [29]. The average NDVI value in the YTRB growing season showed a significant upward trend with a trend rate of 0.011/decade ($p < 0.01$) (Figure 2b). The vegetation coverage in the YTRB was better, and the NDVI value of each growing season was greater than 0.5. The NDVI was prone to saturation in areas with high vegetation coverage and was not sensitive to the increase in vegetation. The vegetation coverage in the YLRB was much lower than that in the YTRB. The average NDVI value in the YLRB also showed a significant upward trend, with a slope of 0.016/decade ($p < 0.01$). During the study period, the average TEM in the YLRB and YTRB was 13.3 °C and 16.3 °C, respectively. The TEM in the growing season of the two basins both increased significantly, with slopes of 0.030 °C/year and 0.039 °C/year ($p < 0.01$), respectively (Figure 2c,d). The average rainfall values in the YLRB and YTRB were 418.3 mm and 901.5 mm, respectively. The accumulated PRE increased at a rate of 0.237 mm/year and 0.921 mm/year, respectively, neither of which passed the significance test. The average TEM and accumulated PRE in the YTRB were significantly higher than those in the YLRB.

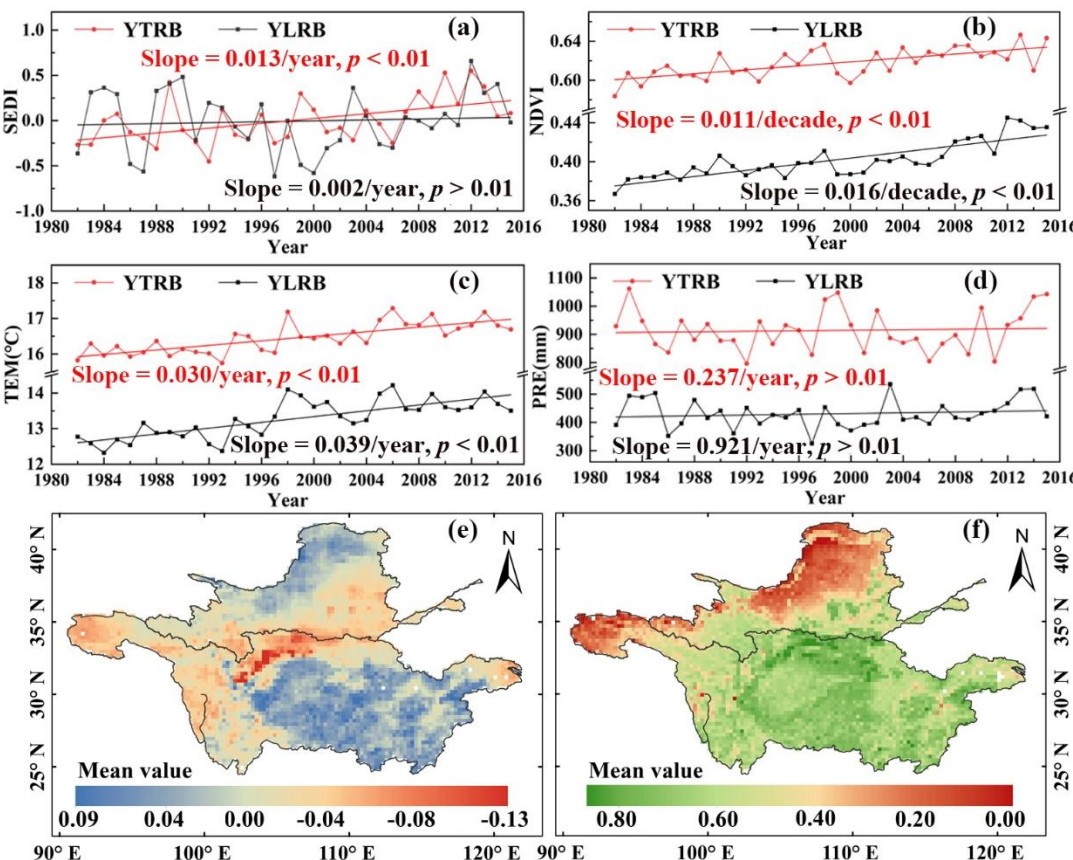

**Figure 2.** Variation of the mean SEDI (**a**), mean NDVI (**b**), mean TEM (**c**), and accumulated PRE (**d**) in the growing season from 1982 to 2015; spatial distribution of the mean SEDI (**e**) and mean NDVI (**f**).

The drought and vegetation coverage in the YLRB and YTRB had clear spatial heterogeneity (Figure 2e,f). The low-SEDI areas (values of less than 0) in the YLRB were mainly distributed in the eastern region, while in the YTRB, the low-SEDI areas were mainly centered in the western region. The NDVI in most areas of the YLRB was lower than 0.5, and was particularly low in the Loess Plateau (<0.2). The vegetation coverage in the southeastern region was slightly higher than that in the middle region of the YLRB. The

vegetation coverage in the YTRB was good, with an NDVI greater than 0.5 in more than 80% of the region. In the central and eastern basins, the vegetation coverage was better than that in other regions. Low-value areas (<0.2) were mainly centered in the upper reaches and were covered by grassland and alpine meadows, as well as glaciers and permanent snow. No significant trend was observed in SEDI in most areas of the YLRB, and the slope remained near 0, thereby maintaining a generally stable state (Figure 3). However, in the YTRB, more than 40% of the areas had a significant trend of wetness, which was mainly centered in the eastern source region and the central and eastern regions. In addition, 9.44% of the area, which was mainly distributed in the northern basin, showed a significant drying tendency (Table 2). According to the annual change trend of vegetation coverage during the growing season, 69.91% of the YLRB showed a significant greening trend and 54.22% of the YTRB area was turning green; these areas were mainly concentrated in the central and eastern regions of both basins.

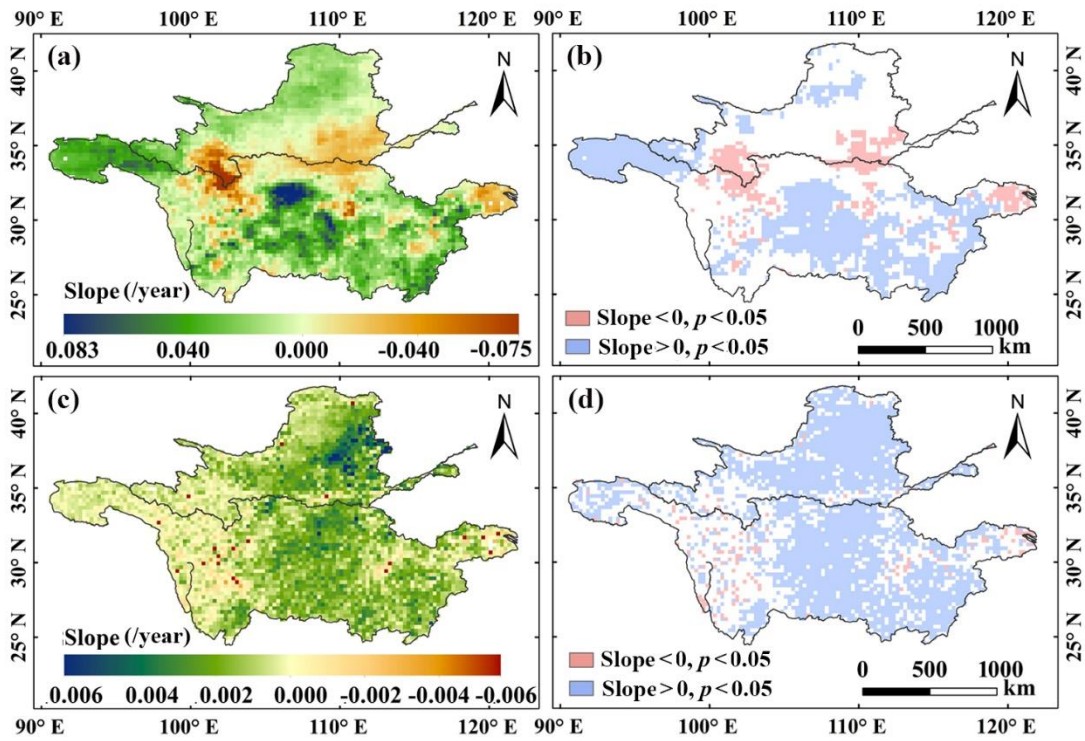

**Figure 3.** Spatial pattern of the slope and significance of the mean SEDI (**a**,**b**) and NDVI (**c**,**d**) of the growing season from 1982 to 2015.

**Table 2.** Proportions of areas with significantly increased and decreased SEDI and NDVI values.

|  | YLRB | | YTRB | |
|---|---|---|---|---|
|  | Significant Increase ($p < 0.05$) | Significant Decrease ($p < 0.05$) | Significant Increase ($p < 0.05$) | Significant Decrease ($p < 0.05$) |
| SEDI | 13.95% | 12.63% | 45.17% | 9.44% |
| NDVI | 69.91% | 1.17% | 54.22% | 4.54% |

### 3.2. Impacts of Drought on Vegetation Change

Figure 4 shows the spatial distribution of the correlation between vegetation coverage and drought in the YLRB and YTRB in the growing season. Drought had a significant impact on vegetation coverage changes, and the impact had obvious spatial heterogeneity. The SEDI value was more negative, indicating that the drought was more serious, which led to a lower NDVI value and inhibition of vegetation growth. In the YLRB, the SEDI and NDVI were

positively correlated in 79.97% of the total area, of which 49.26% passed the 0.05 significance test, which was mainly distributed in the northern basin. Grasslands are widely distributed in these areas, and herbs usually absorb water from the upper soil and respond quickly to changes in rainfall [65]. Because the xylem of herb plants has a low capacity to store water and carbon, it is difficult to withstand the effects of severe drought [66]. In addition, the area was in the Loess Plateau, and the low vegetation coverage and single vegetation type made the ecosystem more fragile. The occurrence of drought events would destroy the stability of the grassland ecosystem and affect the water conditions that vegetation depends on for survival [9,67]. The southwestern basin was also covered with grassland, but its vegetation type was mainly alpine meadow. Because it was covered with snow and ice all year round, the vegetation was less affected by drought [68]. There was a positive correlation between the SEDI and NDVI in 69.34% of the YTRB, and only 20.91% of the regions reached the 95% significance level, indicating that the impact of drought on vegetation in this area was weak. The areas most significantly affected by drought were distributed in the Sichuan Basin, where the main vegetation types were croplands and forests. Soil moisture regulates the surface water and energy cycle by affecting plant growth and crop yield [69]. Croplands are very sensitive to drought, because increasing drought reduces the soil moisture content, which directly affects the cropland vegetation [9]. Although the YTRB is rich in rainfall and vegetation growth can obtain sufficient water under normal conditions, the impact of drought on forests cannot be ignored. Similarly, the study by Chu et al. (2019) [59] showed that under the condition of long-term normal water supply, surface TEM and actual evapotranspiration were still the main factors affecting abnormal forest drought. Severe drought led to changes in the forest canopy structure, such as defoliation and browning of the forest canopy, thereby reducing the vegetation coverage and the NDVI value of the forest. In addition, forests survive by absorbing water from deep-rooted soils during periods of severe drought, and thus, long-term droughts might delay tree growth [70].

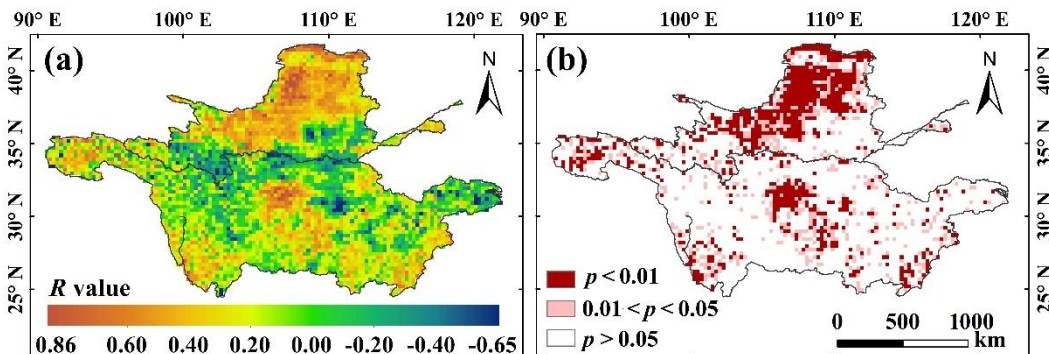

**Figure 4.** Spatial distribution of the correlation (**a**) and significance (**b**) between NDVI and SEDI.

### 3.3. Impacts of Climatic Factors on Vegetation Changes

3.3.1. Correlation Analysis between Climatic Factors and Vegetation Changes

TEM and PRE are the two most important climatic factors leading to drought [14–16], and they are also vital factors affecting vegetation growth [21,59]. The spatial differences in the correlation between TEM, PRE, and vegetation in the YLRB and YTRB were initially explored (Figure 5). In the YLRB, during 1982–2015, the NDVI was positively correlated with TEM in 84.18% of the area, and 32.42% of the region passed the 95% confidence level, which was mainly in the western and southern basin. This was due to the melting of snow and ice under the conditions of warming in the alpine region of the western basin, which promoted the growth of vegetation [71]. PRE in the YLRB had a significant impact on vegetation. The PRE and NDVI were positively correlated in 75.53% of the basin, of which 32.50% of the total area passed the 0.05 significance test, mainly distributed in the northwestern region. In arid and semiarid areas, the amount of rainfall directly affected the variation in the NDVI, but the existence of pioneer vegetation weakened the importance of

PRE in the analysis results [36]. Therefore, the importance of TEM and PRE for vegetation growth varied from region to region. In previous study, the partial correlation between the NDVI and PRE was greater than that of TEM in the YLRB [33]. However, owing to the high altitude and low TEM, the ecological environment of the Qinghai–Tibet Plateau is extremely fragile, and the increase in TEM could promote the growth of vegetation. In contrast, in the YTRB, 60.77% of the total area of the basin showed a significant positive correlation between vegetation coverage and TEM, which was mainly located in the central and eastern regions, indicating that TEM was the key factor leading to vegetation changes in these areas [58]. This result was consistent with the research results of Cui et al. (2020) [60], showing that the vegetation dynamics in the YTRB had a clearer response to TEM than to PRE. TEM and vegetation were insignificantly negatively correlated in the southwestern region of the YTRB. The karst region was widely distributed in these areas, and inadequate rainfall and rising TEM led to a lack of soil moisture and in increase in evapotranspiration, thereby reducing vegetation cover [72]. The relationship between the PRE and NDVI was not significant in most areas. Vegetation growth in humid areas was mainly affected by TEM; PRE was not a key factor in facilitating or limiting the growth of vegetation due to sufficient rainfall in the basin [73].

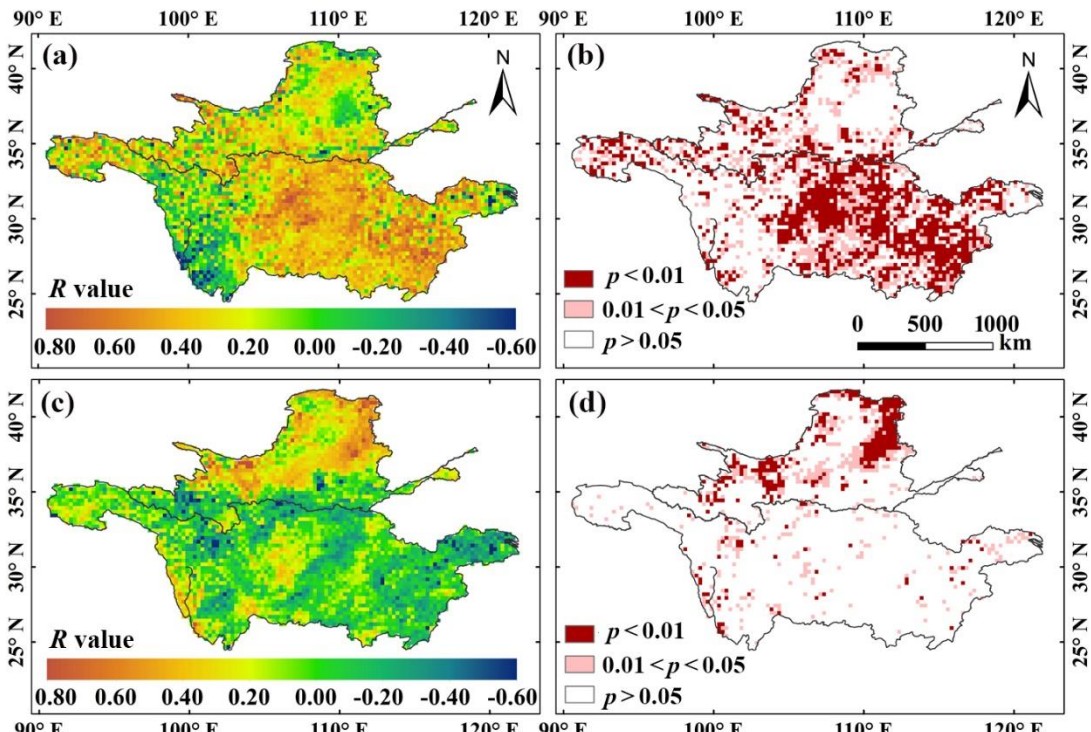

**Figure 5.** Spatial distribution of the correlations and significance between TEM (**a**,**b**), PRE (**c**,**d**), and NDVI.

### 3.3.2. Temporal Effects of Climatic Factors on Vegetation Changes

Because the response of vegetation to climate change was asymmetrical, considering the temporal effects of climatic factors was essential to fully understand the climate–vegetation relationship and to predict the growth of vegetation under global climate change. In this study, the time-lag and time-accumulation effects of TEM and PRE on vegetation were investigated simultaneously, and the best scenario for each pixel was determined. Scenarios that considered both time effects ($R^2$_lagacc) were always better than other scenarios (i.e., only time-lag effects ($R^2$_lag), only time-accumulation effects ($R^2$_acc), or no time effects ($R^2$_no)) in the two basins. Spatially, TEM and PRE in most regions had a significant accumulated effect on vegetation. However, the time-lag effect rarely appeared alone, and it generally appeared at the same time as the time-accumulation effect, i.e., the combined effect. In

addition, these effects showed different results due to differences in vegetation types and climatic factors (Tables 3 and 4).

**Table 3.** Determination coefficients and standard deviations between the NDVI and each climatic factor for the different vegetation types under different scenarios ($R^2$_no, $R^2$_acc, $R^2$_lag, and $R^2$_lagacc).

| | | | Determination Coefficients | | | | Standard Deviations | | | |
|---|---|---|---|---|---|---|---|---|---|---|
| | | | Forests | Grasslands | Croplands | Others | Forests | Grasslands | Croplands | Others |
| YLRB | TEM | $R^2$_no | 0.443 | 0.590 | 0.446 | 0.375 | 0.071 | 0.140 | 0.134 | 0.233 |
| | | $R^2$_acc | 0.499 | 0.694 | 0.510 | 0.462 | 0.071 | 0.132 | 0.147 | 0.252 |
| | | $R^2$_lag | 0.447 | 0.599 | 0.448 | 0.401 | 0.066 | 0.136 | 0.132 | 0.224 |
| | | $R^2$_lagacc | 0.505 | 0.694 | 0.524 | 0.472 | 0.065 | 0.130 | 0.123 | 0.241 |
| | PRE | $R^2$_no | 0.782 | 0.670 | 0.691 | 0.478 | 0.100 | 0.111 | 0.130 | 0.256 |
| | | $R^2$_acc | 0.800 | 0.727 | 0.714 | 0.519 | 0.089 | 0.101 | 0.136 | 0.260 |
| | | $R^2$_lag | 0.786 | 0.693 | 0.695 | 0.497 | 0.091 | 0.095 | 0.129 | 0.247 |
| | | $R^2$_lagacc | 0.800 | 0.727 | 0.715 | 0.522 | 0.089 | 0.101 | 0.133 | 0.256 |
| YTRB | TEM | $R^2$_no | 0.251 | 0.352 | 0.264 | 0.227 | 0.198 | 0.261 | 0.176 | 0.229 |
| | | $R^2$_acc | 0.363 | 0.494 | 0.421 | 0.417 | 0.198 | 0.241 | 0.126 | 0.192 |
| | | $R^2$_lag | 0.327 | 0.435 | 0.354 | 0.339 | 0.161 | 0.210 | 0.116 | 0.178 |
| | | $R^2$_lagacc | 0.395 | 0.521 | 0.441 | 0.433 | 0.165 | 0.209 | 0.109 | 0.182 |
| | PRE | $R^2$_no | 0.472 | 0.562 | 0.604 | 0.582 | 0.276 | 0.235 | 0.156 | 0.172 |
| | | $R^2$_acc | 0.531 | 0.620 | 0.630 | 0.601 | 0.245 | 0.206 | 0.135 | 0.167 |
| | | $R^2$_lag | 0.526 | 0.607 | 0.624 | 0.590 | 0.225 | 0.181 | 0.127 | 0.165 |
| | | $R^2$_lagacc | 0.547 | 0.633 | 0.634 | 0.601 | 0.222 | 0.180 | 0.127 | 0.167 |

**Table 4.** Months and standard deviations of different vegetation types affected by time effects. Acc and Lag indicate time-accumulation and time-lag, respectively.

| | | | Months | | | | Standard Deviations | | | |
|---|---|---|---|---|---|---|---|---|---|---|
| | | | Forests | Grasslands | Croplands | Others | Forests | Grasslands | Croplands | Others |
| YLRB | TEM | Acc | 0.647 | 1.080 | 0.645 | 1.103 | 0.512 | 0.395 | 0.518 | 0.788 |
| | | Lag | 0.103 | 0.516 | 0.112 | 0.552 | 0.306 | 0.502 | 0.316 | 0.502 |
| | PRE | Acc | 1.000 | 1.013 | 0.931 | 1.103 | 0.173 | 0.211 | 0.310 | 0.484 |
| | | Lag | 0.059 | 0.229 | 0.073 | 0.466 | 0.237 | 0.421 | 0.339 | 0.681 |
| YZRB | TEM | Acc | 1.282 | 1.112 | 0.474 | 0.524 | 1.068 | 0.924 | 0.748 | 0.681 |
| | | Lag | 0.758 | 0.585 | 0.209 | 0.181 | 0.996 | 0.873 | 0.533 | 0.455 |
| | PRE | Acc | 1.774 | 1.746 | 1.782 | 2.010 | 1.072 | 0.966 | 0.943 | 0.838 |
| | | Lag | 1.123 | 1.009 | 0.893 | 1.086 | 1.117 | 1.006 | 0.897 | 0.761 |

TEM showed a lag of $0.313 \pm 0.219$ (mean $\pm$ standard deviation) months and an accumulation of $0.872 \pm 0.260$ months in the YLRB. Spatially, TEM showed an accumulation effect of one month on vegetation in 77.94% of the basin (Figure 6a). During the growth and development period of vegetation, not only a suitable TEM level but also a certain amount of heat was required. The accumulation effect of PRE dominated the entire YLRB, and PRE was significantly positively correlated with vegetation in most areas of the basin (Figure 6b). The YLRB is dominated by arid and semiarid areas, so the vegetation growth requires more accumulated water because of the lower monthly rainfall in these areas. In the YTRB, TEM

showed a lag of 0.434 ± 0.372 (mean ± standard deviation) months and an accumulation of 0.938 ± 0.386 months. The grids with no time effect, time-lag, time-accumulation, and combined time effects accounted for 30.16%, 0.82%, 59.12%, and 9.90% of the total grid, respectively. In the Sichuan Basin and the northeastern basin, TEM did not show any time effect, but did have a significant positive correlation with vegetation growth, meaning that the average TEM of the current month continuously affected the vegetation activity [74]. In the source area, TEM showed a one-month accumulation effect on vegetation and no lag effect. The structure and function of the alpine grassland ecosystem in the Qinghai–Tibet Plateau were largely affected by TEM [19,26]. However, the TEM of the current month could not meet the growth needs of vegetation in the cold areas. Only when the accumulation of TEM and heat reached a certain level could vegetation growth be promoted. PRE affected vegetation with an average lag of 0.905 ± 0.353 months and accumulation of 1.691 ± 0.419 months. Spatially, the one-month and two-month accumulated PRE also had a greater impact in the source area and central area of the YTRB, whereas the combined effect was dominant in the southeastern and southwestern regions.

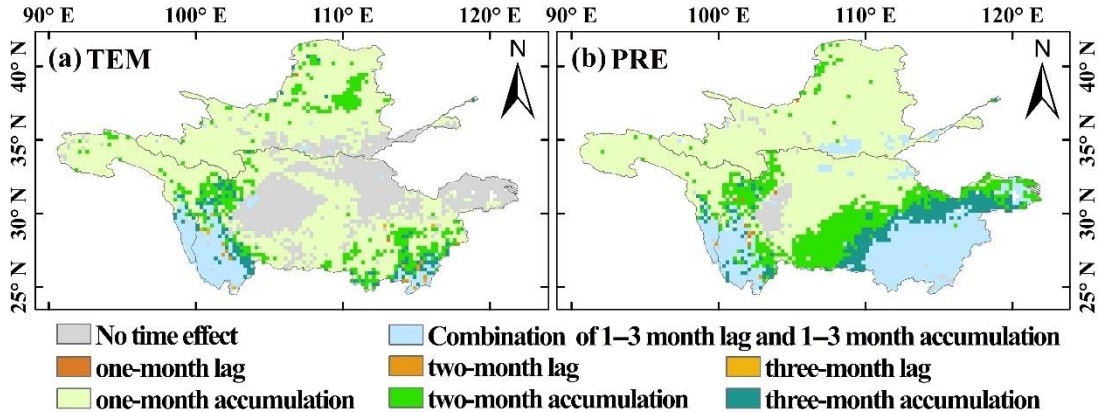

**Figure 6.** Time effects of each climatic factor identified by the maximum determination coefficient.

## 4. Discussion

### 4.1. Comparison of SEDI with Other Drought Indices

Correlation analysis was performed on the SPEI01, SPEI03, SPEI06, SPEI12, SPEI24, and SCPDSI with the SEDI, respectively (Figure 7). Overall, in the YLRB, the SEDI and other indices showed significant positive correlations in almost all pixels. In contrast, the SEDI showed the best correlation with the SPEI on the 6-month timescale; while in the YTRB, relatively poor correlation was found between the SEDI and other drought indices. Previous studies have shown that the SEDI can reasonably capture wet and dry climate variability, especially in arid and semi-arid regions, while in humid and sub-humid regions, the SEDI has large uncertainty due to the small difference between AET and PET, and both are limited by energy [42]. In previous studies, a series of drought indices has been established, such as the standardized precipitation index (SPI), scPDSI, and SPEI. These indices are widely used in drought analysis and monitoring, but they also have some shortages. The SPI is calculated based on precipitation, so it can only reflect atmospheric conditions [75], while the scPDSI fully considers precipitation, soil moisture, runoff, and potential evapotranspiration (PET), but the accumulated errors in its calculation process may lead to large uncertainties in the scPDSI [76,77]. The SPEI can more accurately identify the evolution of drought and its impact on ecosystems by taking into account PET [22,62,78]. Evapotranspiration, an important parameter to quantify drought [79], combines water and energy balances and links the climate system to terrestrial ecosystems [42]. The SEDI was determined by normalizing the evaporative deficit, which was defined as the difference between the AET and PET [40]. Furthermore, Zhang et al. (2019) [42] demonstrated that the drought index considering both actual evapotranspiration (AET) and PET highlighted the intensity of water deficit and the effect on vegetation activity more significantly than the

index on its own, and concluded that the SEDI was more directly related to vegetation water stress than other drought indicators. On the other hand, it is proved that the SEDI was more suitable than general precipitation-based drought indices to study the impacts of drought on crop growth and vegetation productivity because it took into account plant growth and response mechanisms [40]. The SEDI was more practical in highlighting biological effect signals of drought than the drought index based on precipitation and temperature, and had great potential for monitoring drought evolution and the link between drought and ecological responses of terrestrial ecosystems [12,80,81].

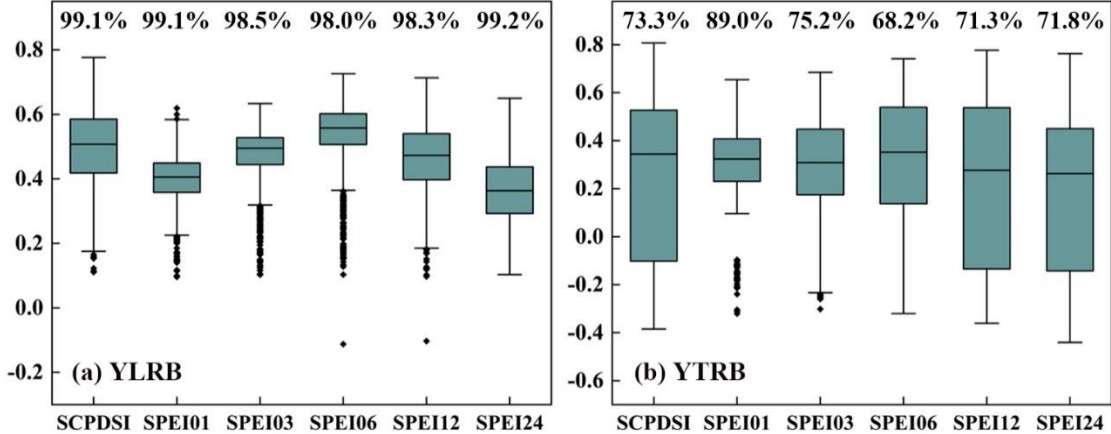

**Figure 7.** Box and whisker plots show the R-values between SEDI and SCPDSI, SPEI01, SPEI03, SPEI06, SPEI12, and SPEI24 for the entire monthly record ($p < 0.05$). The number above the top whisker indicates the percentage of pixels with this correlation of the total pixels in the basin. The thick line in the box represents the median, and the upper and lower parts of the box represent the maximum and minimum values.

### 4.2. Explanation of Vegetation Variation by Climatic Factors

A multiple linear regression model between climatic factors and vegetation changes was established based on the time effects that were determined to best predict the response of vegetation to climate (Table 5). On average, climatic factors have stronger explanatory power for vegetation in the YLRB than in the YTRB, indicating that vegetation in the YLRB was more sensitive to climate change, and that the driving mechanisms for vegetation change in the YTRB were more complicated, except for climatic factors. In the YLRB, climatic factors accounted for 76.7% of the change in the NDVI when taking both time effects into account. The explanatory power of the accumulated effect was greater than that of only the time-lag effect, which were 76.6% and 73.6%, respectively. Among the different vegetation types, 74% to 81% of the change in the NDVI was explained by climatic factors. Compared with scenarios that do not consider time effects, the explanatory power of climatic factors on the NDVI with time effects increased by 5.3%, and the percentage that could be explained in forests, grasslands, and croplands increased by 2.5% to 6.3%. In the YTRB, when all time effects were considered, climatic factors explained 63.2% of the change in vegetation. Among the different vegetation types, 55–66% of the change in the NDVI was explained by climatic factors. Compared with models that did not consider time effects, the explanation of climatic factors on the vegetation changes with time effects increased by 4.8%, and the percentage that could be explained in forests, grasslands, and croplands increased by 2.4% to 5.7%. For different vegetation types, the accumulated effect of climatic factors over time contributed more to vegetation growth than the time-lag effect. This might be due to the complex nonlinear threshold of the vegetation response. Individual plants require the accumulation of PRE and TMP to start the plant life cycle [74,82]. The biogeochemical cycle that provided soil nutrients for plants to grow was also subject to time-accumulation effects [83]. Therefore, the consideration of temporal effects was helpful in improving the predictability of vegetation growth affected by climatic factors.

**Table 5.** Determination coefficients of multiple linear regression models for different vegetation types in different scenarios.

| | YLRB | | | | YTRB | | | |
|---|---|---|---|---|---|---|---|---|
| | Forests | Grasslands | Croplands | Others | Forests | Grasslands | Croplands | Others |
| $R^2$_no | 0.711 | 0.788 | 0.723 | 0.710 | 0.614 | 0.505 | 0.597 | 0.619 |
| $R^2$_acc | 0.763 | 0.806 | 0.786 | 0.732 | 0.657 | 0.544 | 0.648 | 0.640 |
| $R^2$_lag | 0.734 | 0.793 | 0.749 | 0.716 | 0.639 | 0.535 | 0.631 | 0.633 |
| $R^2$_lagacc | 0.764 | 0.807 | 0.786 | 0.735 | 0.662 | 0.552 | 0.654 | 0.643 |

Ding et al. (2020) [61] used the same method to investigate the time effects of TEM, PRE, and radiation in the growing season on vegetation growth globally, and similar conclusions were obtained. Spatially, in the YLRB and YTRB, TEM had an accumulated effect or no time effect, and PRE had a one-month and two-month accumulated effect on vegetation. Similarly, Wu et al. (2015) [27] only focused on the time-lag effect of climate change on global vegetation, and their results demonstrated that vegetation growth in the mid–high latitudes of the northern hemisphere had the greatest correlation with the TEM of the same month without obvious time-lag effect. They believed that the vegetation growth was not determined by the PRE of the current month, but rather by the PRE of the previous months together in arid and semiarid regions. In our study, the accumulated effect of PRE on vegetation clearly exceeded the lag effect, indicating that the accumulated PRE in the previous months could better meet the water demand for vegetation growth in the arid and semiarid areas of the YLRB. Wen et al. (2019) [84] found that the interpretation and fitting of climatic factors to vegetation changes significantly improved after considering the time-accumulation effects of climate. Long et al. (2010) [85] pointed out that the accumulated PRE in the previous months and the current month seems to be a better explanation for the growth of grassland vegetation in Inner Mongolia. According to Ivits et al. (2016) [83], consideration of accumulated PRE could more accurately assess the stability of ecosystems against drought. Therefore, we believed that the accumulated effect cannot be ignored in research on the influence of climate on vegetation in arid and semiarid regions. In summary, time-lag and time-accumulation effects had complex spatial patterns and varied by region and vegetation type. These results not only deepen our understanding of the relationship between climate and vegetation growth in the YLRB and the YTRB, but also have reference significance for managing and predicting vegetation on a regional scale.

*4.3. Contributions of Anthropogenic Factors to Vegetation Variation*

Changes in vegetation that cannot be explained by climate change might be caused by anthropogenic factors [4,86,87], such as the rapid development of urbanization [88], the implementation of large numbers of ecological projects [89], and the migration of populations [85]. The residual trend method was adopted to evaluate the impact of human activities on vegetation dynamics, which showed significant spatial differences (Figure 8). In general, the positive contribution of human activities in the YLRB was greater than the negative contribution [33], and human activities in the eastern part of the YLRB had a significant positive impact on vegetation. The residuals in the central and eastern regions of the YLRB showed a significant increasing trend. Residuals in high-altitude areas such as the Qinghai–Tibet Plateau in the western basin showed a decreasing trend. This area provided a good habitat for vegetation growth, but the negative impact of human activities exceeded the carrying capacity, and it was difficult to maintain an increase in the NDVI. Previous researchers have also paid great attention to the effects of human activities on vegetation changes in the YLRB. Yi et al. (2014) [90] proved that the project of returning farmland to forest and grassland greatly promoted the increasing of the NDVI in the Loess Plateau, whereas urban expansion, deforestation, and overgrazing led to a decrease in the NDVI. Liu et al. (2021) [32] believed that the impact of human activities on vegetation changes

was gradually deepening, and ecological construction projects played a significant role in promoting vegetation restoration. Zhang et al. (2021) [91] concluded that the average contribution rate of human activities to vegetation changes during the 1982–2015 growing season was 69% in the YLRB, and it showed clear seasonal and regional differences. In the YTRB, the regions with significant increases in residuals were distributed in the middle, southwest, and southeast areas of the basin, while the residuals showed a downward trend in the western alpine region and the eastern Yangtze River Delta. This coincided with the results of Qu et al. (2020) [38], who pointed out that the areas with increased forest area (mainly in the central region) had a strong spatial consistency with the areas where human activities had a positive impact on vegetation. This might be directly related to the implementation of large-scale natural forest resource protection projects, nature reserve construction projects, Yangtze River Shelterbelt construction projects, and the Conversion of Cropland to Forest Project [2]. Moreover, with the rapid development of urbanization, large amounts of croplands have been converted to urban land, such as the Yangtze River Delta region [31], and population migration is also constantly occurring. The rapid increase in population greatly accelerated the intensity and speed of land development, the area of urban construction land increased, and the migration of large-scale working-age people caused a large area of arable land to be unused or degraded into grassland, thereby leading to a decrease in the regional NDVI [58,92].

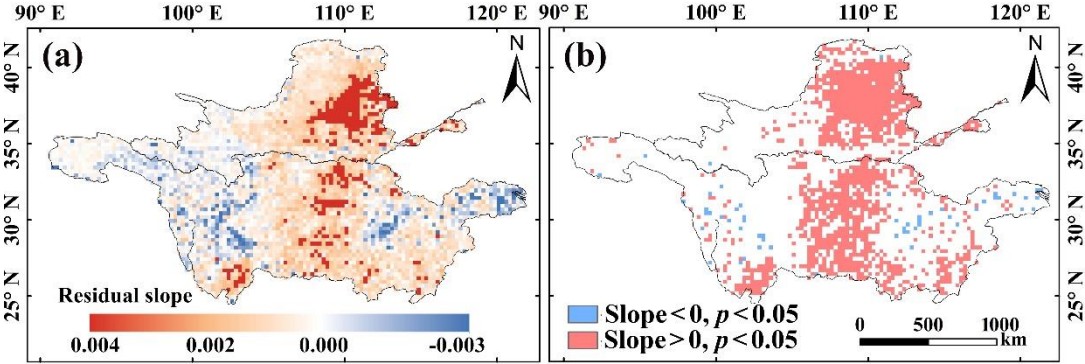

**Figure 8.** Spatial distribution of the NDVI residual variation trend (**a**) and significance (**b**).

*4.4. Limitations*

This study has some limitations and uncertainties. Firstly, other climatic factors (such as solar radiation and soil moisture), non-climatic factors (such as carbon dioxide fertilization and nitrogen deposition), and natural disturbances (such as wildfires and insect pests) also have a non-negligible impact on vegetation growth [38,43,87]. Furthermore, although this research initially discussed the contribution of human activities, a detailed and quantitative impact analysis on the types of human activities was not conducted. In addition, the residual analysis was based on the assumption that the impact of climate on vegetation is linear. This assumption was subjective. In the future, it will be necessary to establish a more scientific and rigorous model or method to separate the impacts of climate and human activities on vegetation.

## 5. Conclusions

In this study, Sen's slope and the MK trend test were used to analyze the temporal and spatial changes in vegetation coverage and drought during the growing season in the YLRB and YTRB from 1982 to 2015. The relationships between drought and vegetation, and climatic factors (TEM and PRE) were then investigated using Pearson correlation analysis. The effects of TEM and PRE on vegetation were analyzed by considering the time-lag and time-accumulation effects of 1–3 months. The contribution of anthropogenic factors to vegetation dynamics was also discussed using the residual trend method. The main findings are summarized as follows:

From 1982 to 2015, the average SEDI in the YLRB fluctuated greatly without a clear variation trend, but that in the YTRB significantly increased at a rate of 0.013/year ($p < 0.01$). Spatially, the variation trend of the SEDI in most areas of the YLRB was not obvious, while in YTRB, more than 40% of the area showed a significant trend of wetness, which was mainly in the eastern source area and the central and eastern regions. The NDVI of the two basins significantly increased with growth rates of 0.011/decade and 0.016/decade ($p < 0.01$), respectively. Vegetation coverage was the lowest in the Qinghai–Tibet Plateau and the Loess Plateau.

The effects of drought on vegetation coverage exhibited clear spatial heterogeneity. In the YLRB, the SEDI and NDVI were significantly positively correlated in 49.26% of the regions, which were mainly concentrated in the northern basin. This was related to the growth characteristics of grasslands and the low complexity of vegetation types. In the YTRB, the vegetation was weakly affected by drought, and only 20.91% of the area had a significant positive correlation between the SEDI and NDVI. The area most severely affected by drought was distributed in the Sichuan Basin, which was attributed to drought, leading to a decrease in soil moisture, directly affecting the growth of vegetation in croplands.

The NDVI in the upper and middle reaches of the YLRB was significantly positively correlated with TEM, whereas PRE in the Loess Plateau played a vital role in vegetation growth. The one-month accumulated effect of TEM and PRE on vegetation was dominant. The response of vegetation dynamics to TEM was greater than that of PRE in the YTRB. TEM was the main factor leading to changes in vegetation, especially in the central and eastern regions. In the source area, TEM had a one-month accumulated effect on vegetation, indicating that accumulated TEM could promote the growth of vegetation in cold regions.

Taking the time-lag and time-accumulation effects into account, climatic factors explained 76.7% and 63.2% of the vegetation changes in the YLRB and YTRB, respectively. The explainable percentage changes of different vegetation types in the two basins increased by 2% to 6%, respectively, and the time-accumulation effect of climatic factors had a stronger explanatory power for vegetation growth than the time-lag effect. The impact of anthropogenic factors on vegetation was also not negligible.

**Author Contributions:** W.J., Z.N. and L.W. designed the research; W.J. and F.X. performed the experiments and analyzed the data; W.J. wrote the manuscript; R.Y., X.G. and Y.J. revised the manuscript. All authors have read and agreed to the published version of the manuscript.

**Funding:** This work was financially supported by the National Natural Science Foundation of China (41925007, 41771360, 41975044, and 41801021), and the Fundamental Research Funds for National Universities, China University of Geosciences, Wuhan.

**Institutional Review Board Statement:** Not applicable.

**Informed Consent Statement:** Not applicable.

**Data Availability Statement:** The standardized evapotranspiration deficit index was provided by DIGITAL.CSIC (https://digital.csic.es/handle/10261/160091, accessed on 20 February 2021). The Normalized Difference Vegetation Index (NDVI) was obtained from the Global Inventory Monitoring and Modeling Systems (GIMMS) group (https://ecocast.arc.nasa.gov/data/pub/gimms/, accessed on 20 February 2021). The CN05.1 meteorological dataset was provided by the Laboratory for Climate Studies of the China Meteorological Administration. The land-use data were derived from the MCD12Q1 in 2015 (https://ladsweb.modaps.eosdis.nasa.gov/, accessed on 24 March 2021). The SPEI and scPDSI datasets were downloaded from http://digital.csic.es/handle/10261/153475 and https://crudata.uea.ac.uk/cru/data/drought, respectively (accessed on 15 January 2022).

**Acknowledgments:** We are grateful to professor Wu Jia of the China Meteorological Administration for providing the CN05.1 meteorological dataset.

**Conflicts of Interest:** The authors declare no conflict of interest.

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
