# Peer review of "Impacts of Drought and Climatic Factors on Vegetation Dynamics in the Yellow River Basin and Yangtze River Basin, China"

_remotesensing, doi:10.3390/rs14040930_

Round 1

Reviewer 1 Report

The study looked into the long-term drought characteristics over the Yellow River Basin and Yangtze River Basin as well as factors that both affected the evolution of the droughts and the ones that drought affected. I particularly appreciate the seriousness with which the study was carried out and the writing was done. Furthermore, the section on “limitations” is certainly important and shows the foresight of the authors which is rather missing from many modern studies.

I have a few comments that I hope will make the MS clear for the readership:

  1. The choice of SEDI for the study seems relevant for the aim, but I wonder what extra information we are learning from SEDI that we have not learned from SPEI. I know that could be an entire work on its own, but I am sure some discussions could be included in the DISCUSSION section. See studies like (10.1002/2016JD026168) which does a comprehensive review of drought indices over China.
  2. NDVI, which basically describes the greenness of vegetation, is a good choice as a proxy for vegetation. Nonetheless, studies have shown that biomass based on datasets such as VOD (vegetation optical depth) may better characterize vegetation water which is essential for drought (see https://www.nature.com/articles/nclimate2581/). Perhaps a little discussion on the advantages of using NDVI might be good when introducing the NDVI data.
  3. The pearson correlation has been a very good analysis tool for a long time but unfortunately, it does not bear the needed asymmetry needed to infer causation. Correlation mainly quantify the similarities between two centered time series. I think you could rephrase lines 172-173.
  4. Rightly, the lag periods chosen are also noted in previous studies. However, the problem is that stopping at 3-months lag doesn’t guarantee that it stops at that lag. Please include some discussions should be justify this.
  5. The left figure of Figure 2 has (-0.00). Please correct it.
  6. How do you determine whether the impact of droughts on the factors and the impact of he factors on drought. It seems they are mixed up here
  7. Around line 378, the authors quote a percentge of change ndvi. How were these percentages computed?

Best regards

Author Response

Reviewer 1

--- Query #1 ---

The choice of SEDI for the study seems relevant for the aim, but I wonder what extra information we are learning from SEDI that we have not learned from SPEI. I know that could be an entire work on its own, but I am sure some discussions could be included in the DISCUSSION section. See studies like (10.1002/2016JD026168) which does a comprehensive review of drought indices over China.

--- Answer #1 ---

Thank you very much for your suggestions, we have added it. (see Page 11, Section 4.1)

4.1 Comparison of SEDI with other drought indices

Correlation analysis was performed on SPEI01, SPEI03, SPEI06, SPEI12, SPEI24, and SCPDSI with SEDI, respectively (Figure 7). Overall, in the YLRB, SEDI and other indices showed significant positive correlations in almost all pixels, and the SEDI has the best correlation with SPEI on the 6-month timescale; while in the YTRB, relatively poor correlation was showed between SEDI and other drought indices. Previous studies have shown that SEDI can reasonably capture wet and dry climate variability, especially in arid and semi-arid regions, while in humid and sub-humid regions, SEDI have large uncertainty due to the small difference between AET and PET and both are limited by energy [1].

In previous studies, a series of drought indices has been established, such as the standardized precipitation index (SPI), scPDSI, and SPEI. These indices are widely used in drought analysis and monitoring, but they also have some shortages. SPI is calculated based on precipitation, so it can only reflect atmospheric conditions [2], while scPDSI fully considers precipitation, soil moisture, runoff, and potential evapotranspiration (PET), but the accumulated errors in its calculation process may lead to large uncertainties in scPDSI [3, 4]. SPEI can more accurately identify the evolution of drought and its impact on ecosystems by taking into account PET [5-7]. Evapotranspiration, an important parameter to quantify drought [8], combines water and energy balances and links the climate system to terrestrial ecosystems [1]. SEDI is determined by normalizing the evapotranspiration deficit, which is defined as the difference between AET and PET [9]. Furthermore, Zhang et al. (2019) [1] demonstrated that the drought index considering both actual evapotranspiration (AET) and PET highlighted the intensity of water deficit and the effect on vegetation activity more significantly than the index based on its own, and they concluded that SEDI was more directly related to vegetation water stress than other drought indicators. On the other hand,it is proved that SEDI was more suitable than general precipitation-based drought indices to study the impacts of drought on crop growth and vegetation productivity because it takes into account plant growth and response mechanisms [9]. SEDI was more practical in highlighting biological effect signals of drought than the drought index based on precipitation and temperature, and had great potential for monitoring drought evolution and the link between drought and ecological responses of terrestrial ecosystems [10-12].

--- Query #2 ---

NDVI, which basically describes the greenness of vegetation, is a good choice as a proxy for vegetation. Nonetheless, studies have shown that biomass based on datasets such as VOD (vegetation optical depth) may better characterize vegetation water which is essential for drought (see https://www.nature.com/articles/nclimate2581/). Perhaps a little discussion on the advantages of using NDVI might be good when introducing the NDVI data.

--- Answer #2 ---

Thank you very much for your suggestions, we have added it. (see Section 2.2, Page 4, Line 151-159).

NDVI has become the most widely used vegetation evaluation index due to its long history, simplicity and dependence on easily obtained multi-spectral bands [13]. It is closely correlated with a series of interrelated biomass variables such as leaf area index [14], vegetation cover [15] and green biomass [16]. NDVI is an effective tool for coupling climate and vegetation distribution and performance on large spatio-temporal scales [17]. NDVI analysis of contemporary vegetation dynamics is particularly effective for large transboundary geographical regions with varied terrain, diverse vegetation and diverse land use types in the Northern Hemisphere [18].

--- Query #3 ---

The Pearson correlation has been a very good analysis tool for a long time but unfortunately, it does not bear the needed asymmetry needed to infer causation. Correlation mainly quantify the similarities between two centered time series. I think you could rephrase lines 172-173.

--- Answer #3 ---

Thank you very much for your suggestions, we have rephrased it. (see Section 2.3.2, Page 5, Line 184-187)

Pearson correlation analysis is a widely used method in statistics, has been widely used in previous studies [19-21]. In this study, this method was used to examine the correlation between SEDI and NDVI during the growing season, as well as be-tween climatic factors and NDVI.

--- Query #4 ---

Rightly, the lag periods chosen are also noted in previous studies. However, the problem is that stopping at 3-months lag doesn’t guarantee that it stops at that lag. Please include some discussions should be justify this.

--- Answer #4 ---

Thanks for your careful reading of our manuscript. First of all, the purpose of our research is not to explore the longest time-lag effect of temperature and precipitation on vegetation, but to explore the dual time-lag effect and cumulative effect of temperature and precipitation within a reasonable range on vegetation in the Yellow and Yangtze River basins. And then, previous studies have shown that the time effect on the monthly scale is usually less than 3 months [6, 22, 23],, so it is valuable to study the lag effect in the range of 0-3 months. Besides, in Section 4.2 of DISCUSSION, it is mentioned that: Ding et al. (2020) [22] used the same method to study the temporal effects of growing season TEM, PRE and solar radiation on global vegetation growth. Spatially, in the Yangtze River and Yellow River basins, TEM had no obvious time-lag effect on vegetation, while PRE was dominated by 1-month and 2-month cumulative effects. Wu et al. (2015) [23] studied only the time-lag effect of climate change on global vegetation, and the results showed that the vegetation growth in the middle and high latitudes of the northern hemisphere had the greatest correlation with the monthly TEM, but there was no obvious time-lag effect. They argued that vegetation growth in arid and semi-arid regions is not determined by the PRE of the current month, but by the PRE of previous months together, so it is necessary to consider the cumulative effect. Finally, in fact, our research results also showed that in the Yangtze and Yellow River basins, the lag effect of temperature and precipitation on vegetation is not significant, and the cumulative effect of temperature and precipitation on vegetation change is significantly greater than their lag effect.

--- Query #5 ---

The left figure of Figure 2 has (-0.00). Please correct it.

--- Answer #5 ---

Thank you very much for your suggestions, we have corrected and revised it. (see Page 6, Figure 2)

--- Query #6 ---

How do you determine whether the impact of droughts on the factors and the impact of the factors on drought. It seems they are mixed up here.

--- Answer #6 ---

Thanks for your careful reading of our manuscript. In this study, firstly, the Pearson correlation analysis was adopted to study the impact of drought on vegetation. Then, since temperature and precipitation are the key factors causing droughts and are also important factors affecting vegetation growth, a preliminary study was conducted on the influence of climatic factors on vegetation by using correlation analysis. And then, the time effect (including lag effect and cumulative effect) of the two climatic factors on vegetation was deeply discussed by the method of linear regression. There is no doubt that severe drought events can have devastating effects on vegetation growth [24-26]. Temperature and precipitation are important climatic factors for the formation of drought, which has also been confirmed in previous studies [27-29]. On the other hand, numerous studies have shown that air temperature and precipitation are important factors affecting vegetation dynamics [19, 30]. In previous studies, Pearson correlation analysis was widely used to study the impact of drought on vegetation [31-34] and the impact of climate change on vegetation dynamics [35-37] at the global and regional scales, demonstrating the feasibility and applicability of this approach in this study and similar studies.

--- Query #7 ---

Around line 378, the authors quote a percentage of change NDVI. How were these percentages computed?

--- Answer #7 ---

Thanks for your careful reading of our manuscript. The determination coefficient (R2) of multiple linear regression model can represent the proportion of the total variation of dependent variables that can be explained by independent variables in the regression model, and is one of the indicators to measure the effect of the established model. In this study, with NDVI as the dependent variable and TEM and PRE under different time effects as independent variables, multiple linear regression models of NDVI, TEM and PRE for each pixel were established at the grid scale to obtain R2 under different conditions. Then the mean value of R2 is calculated and expressed as a percentage on the basin scale, which can more intuitively understand the proportion of TEM and PRE explanations in NDVI variation.

Reviewer 2 Report

Please find my review of a manuscript titled "Impacts of drought and climatic factors on vegetation dynamics in the Yellow River Basin and Yangtze River Basin, China " by Weixia Jiang, Zigeng Niu, Lunche Wang, Rui Yao, Xuan Gui, Feifei Xiang, Yuxi Ji submitted for consideration for possible publication in MDPI Remote Sensing.

This study investigates the impacts of drought and climate change on vegetation dynamics in two river basins of China. Specifically, the correlations between drought, climatic factors and vegetation conditions, as well as the time-lag and time-accumulation effects of climatic factors on vegetation coverage were investigated. The standardized evapotranspiration deficit index (SEDI) was selected to quantify the severity of drought. SEDI is an appropriate index for studying the impact of drought on vegetation. The Normalized Difference Vegetation Index (NDVI) data obtained from the Global Inventory Modeling and Mapping Study (GIMMS 3 g) was used as the vegetation index. This dataset spans from 1982 to 2015 and having such long time series is beneficial for climate studies.

The subject of this study is suitable for Remote Sensing journal. Data and methodology are robust. Results could be used for formulating vegetation management strategies and predicting future vegetation growth.

Some revision is required before publishing the manuscript.

Abstract.

Lines 23-24. The NDVI of the two basins increased significantly at rate of 0.011/decade and 0.016/decade, respectively (p<0.01).

Clarification is required, e.g., The NDVI of the two basins, YLRB and YTRB, increased significantly at rate of 0.011/decade and 0.016/decade, respectively (p<0.01). 

Lines 24-26. Drought had a significant impact on vegetation in 49.26% of the YLRB area, which was mainly located in the northern region. In the YTRB, the area significantly affected by drought accounted for 20.91%, which was mainly distributed in Sichuan Basin.

The authors may wish to consider replacing 49.26% and 20.91% with 49% and 21%, unless they wish to clarify why precision to two decimal places is important to correctly present their results.

Lines 27-29. In the YLRB, both generally had a one-month accumulated effect on vegetation conditions, while in the YTRB, temperature was the major factor leading to changes in vegetation. 

Clarification is required, e.g., both temperature and precipitation.

  1. Materials and Methods

2.1. Study area. Figure 1b is referenced on Line 129. Reference for Figure 1a is missing.   

Line 151. The monthly PRE and TEM …

Start a new paragraph with this sentence.

  1. Results.

Lines 211-212. Due to the lack of PRE, drought events occurred frequently [59]. 

Clarification is required, e.g., in YLRB.

Lines 213-215. The average NDVI value in the YTRB growing season of showed a significant upward trend with a trend rate of 0.011/decade (p<0.01) (Figure 2b). 

Delete "of" after "season".

Line 292. 3.1.1. 3.3.1 Correlation analysis between climatic factors and vegetation changes.

Delete 3.1.1.

General comment: as English is not the first language of the authors, I recommend using MDPI editing service for improving quality of this manuscript.

This reviewer recommends accepting the manuscript after suggested minor revision.

Yours faithfully,

The Reviewer

Author Response

Reviewer 2

--- Query #1 ---

Lines 23-24. The NDVI of the two basins increased significantly at rate of 0.011/decade and 0.016/decade, respectively (p<0.01).

Clarification is required, e.g., The NDVI of the two basins, YLRB and YTRB, increased significantly at rate of 0.011/decade and 0.016/decade, respectively (p<0.01).

--- Answer #1 ---

Thank you very much for your suggestions, we have revised it.

Thanks for your careful reading of our manuscript, we made our best efforts to rewrite. (see Page 1, Line 23-24)

--- Query #2 ---

Lines 24-26. Drought had a significant impact on vegetation in 49.26% of the YLRB area, which was mainly located in the northern region. In the YTRB, the area significantly affected by drought accounted for 20.91%, which was mainly distributed in Sichuan Basin.

The authors may wish to consider replacing 49.26% and 20.91% with 49% and 21%, unless they wish to clarify why precision to two decimal places is important to correctly present their results.

--- Answer #2 ---

Thank you very much for your suggestions, we have revised it. (see Page 1, Line 24-26)

--- Query #3 ---

Lines 27-29. In the YLRB, both generally had a one-month accumulated effect on vegetation conditions, while in the YTRB, temperature was the major factor leading to changes in vegetation.

Clarification is required, e.g., both temperature and precipitation.

--- Answer #3 ---

Thanks for your careful reading of our manuscript, we are so sorry for vague expression. (see page 1, Line 27-29)

--- Query #4 ---

2.1. Study area. Figure 1b is referenced on Line 129. Reference for Figure 1a is missing.  

Line 151. The monthly PRE and TEM … Start a new paragraph with this sentence.

--- Answer #4 ---

Thank you very much for your suggestions, we have revised it. (see Page 3, Line 115 and Line 121; Page 4, Line 164)

--- Query #5 ---

Lines 211-212. Due to the lack of PRE, drought events occurred frequently [59].

Clarification is required, e.g., in YLRB.

--- Answer #5 ---

Thanks for your careful reading of our manuscript, we have revised it. (see Page 6 Line 220-222) 

--- Query #6 ---

Lines 213-215. The average NDVI value in the YTRB growing season of showed a significant upward trend with a trend rate of 0.011/decade (p<0.01) (Figure 2b).

Delete "of" after "season".

--- Answer #6 ---

Thanks for your careful reading of our manuscript, we have deleted it. (see Page 6 Line 223-225)

--- Query #7 ---

Line 292. 3.1.1. 3.3.1 Correlation analysis between climatic factors and vegetation changes.

Delete 3.1.1.

--- Answer #7 ---

Thank you for reading our manuscript carefully, we have deleted it. (see Page 8, Line 301) 

--- Query #8 ---

General comment: as English is not the first language of the authors, I recommend using MDPI editing service for improving quality of this manuscript.

--- Answer #8 ---

Thank you very much for your constructive suggestions. Before our submission, we have conducted editing service for improving quality of this manuscript in other professional institutions.

Round 2

Reviewer 1 Report

I thank the authors for addressing my comments. 

All the best!